# Latent Tuberculosis Infection among Health Workers in Germany—A Retrospective Study on Progression Risk and Use of Preventive Therapy

**DOI:** 10.3390/ijerph18137053

**Published:** 2021-07-01

**Authors:** Nika Zielinski, Johanna Stranzinger, Hajo Zeeb, Jan Felix Kersten, Albert Nienhaus

**Affiliations:** 1Faculty 11, Human and Health Sciences, University of Bremen, 28359 Bremen, Germany; nika.zielinski@bgw-online.de (N.Z.); zeeb@bips.uni-bremen.de (H.Z.); 2Department of Occupational Medicine, Toxic Substances, Health Service Research, German Statuary Institution for Accident Insurance and Prevention for Health and Welfare Services (BGW), 22089 Hamburg, Germany; johanna.stranzinger@bgw-online.de; 3Leibniz Institute for Prevention Research and Epidemiology—BIPS, 28359 Bremen, Germany; 4Competence Centre for Epidemiology and Health Services Research for Healthcare Professionals (CVcare), Institute for Health Services Research in Dermatology and Nursing (IVDP), University Medical Centre Hamburg-Eppendorf (UKE), 20246 Hamburg, Germany; j.kersten@uke.de

**Keywords:** latent tuberculosis infection (LTBI), progression, Tuberculosis Preventive Therapy (TPT), health workers, occupational health

## Abstract

Despite the decline in tuberculosis incidence (TB) in Germany, health workers (HW) are at greater risk of becoming infected with *Mycobacterium tuberculosis*. To date, little is known about the risk of progression of latent tuberculosis infections (LTBI) and the use of Tuberculosis Preventive Therapy (TPT) among HW. Routine data from the German Statutory Institution for Accident Insurance and Prevention for Health and Welfare Services (BGW) were analysed and a retrospective survey was conducted. A self-administered questionnaire was sent to 1711 HW who had received recognition of an LTBI as an occupational disease between the years 2009 and 2018. The response rate was 42.3% after correcting for those with no actual address (20.4%). We included 575 HW in the data analysis of the retrospective survey. The cumulative incidence of progression, the incidence density and the associated 95% confidence interval (95% CI) were calculated. Three progressive cases were identified in the analysis of the routine data. In the survey cohort, three HW developed TB during the observation period of 5.4 years on average (standard deviation: 2.8 years; interquartile range: 5.0 years). The cumulative TB incidence was 0.52% in the survey group (95% CI: 0.14% to 1.65%). The incidence density was 0.97 cases per 1000 person years (95% CI: 0.25 to 3.10). One-third of the respondents underwent TPT. Significant differences were observed between age and activity groups in the use of TPT, but not between the genders, year of diagnosis or the reason for performing the screening. The data indicate that the risk of progression of an LTBI is low for HW. However, one-third of the HW had undergone TPT. Information about the expected progression risk is important so that it can be weighed against the risk of side effects of TPT.

## 1. Introduction

Health workers (HW) are at greater risk of infection compared to non-medical occupational groups due to their occupational exposure to tuberculosis (TB) [1]. In Germany, the prevalence of Latent TB Infection (LTBI) among health workers is estimated to be around 10% based on examinations conducted in connection with preventative occupational medicine [2,3]. TB infections and illnesses of occupational origin may be recognised as occupational diseases in workers in healthcare, welfare, laboratories or in positions where there is an elevated risk of infection [4]. For the German Statutory Institution for Accident Insurance and Prevention for Health and Welfare Services (BGW), around half of all recognised occupational diseases caused by infection (BK 3101) are attributable to LTBI and TB cases [5]. 

COVID-19 and TB are both airborne diseases. As the infection-risk of HW for TB is well established, there can be lessons learnt from TB prevention for COVID-19 prevention. However, it needs to be emphasised that the number of HW with COVID-19 is much higher than the number of HW with LTBI as an occupational disease. In Germany, during the last 13 months, more than 50,000 occupational diseases because of COVID-19 were recognised, while about 200 occupational diseases because of LTBI are recognised every year. 

After infection with *Mycobacterium tuberculosis*, it is possible that the disease may progress to active TB, requiring treatment. According to estimates of the World Health Organization (WHO), 5 to 10% of infected adults have an LTBI that progresses to active TB during lifetime [6]. The LTBI may be treated with Tuberculosis Preventive Therapy (TPT) to potentially reduce the risk of progression by up to 90% [7]. TPT aims to eliminate bacteria during its dormant (“inactive”) state [8]. It is necessary to keep the progression rate among HW low in the interest of infection and patient safety [9]. 

Little is known about the progression risk of LTBI among HW to date. In Germany, no cases with progression were found among HW in two studies [10,11]. However, these studies did not systematically examine progression risk. In Portugal, a progression rate among HW of 0.4% was found in two years [12]. No studies could be found on the use of TPT among HW in Germany. This study aims to gain new insights into the progression rate among HW and to estimate how many undergo TPT. 

## 2. Materials and Methods

The analysis is based on a routine dataset from BGW, which includes all LTBI occupational disease cases recognised from 2009 until 2018. A total of 1711 cases of occupation-related LTBI were recognised in this time period (Table 1), which are on average 171 cases per year with no obvious time trend. To estimate progression rates among health workers, two strategies were pursued.

First, BGW’s routine data were examined for progressive cases. This involved pairing LTBI and TB cases using personal identification numbers for all recognised occupational diseases arising from an LTBI between 2009 and 2018. This enabled the identification of insured patients for whom an LTBI had first been reported and subsequently a second file had been opened for TB as an occupational disease. A case file analysis was conducted of high-cost LTBI cases, because it is also possible for the TB to be registered in connection with a previously recognised occupational disease (in this case LTBI). In 2012, Diel et al. [13] calculated the average direct costs of standard tuberculosis treatment to be EUR 7363.99. For this reason, case files were requested for insurance claims involving LTBI and treatment costs of EUR 7000.00 or more. These were analysed manually in relation to a progression.

For an LTBI to be recognised as an occupational disease, different requirements need to be fulfilled. The LTBI must be confirmed by a positive Interferon Gamm Release Assay (IGRA) and TB must be excluded by chest X-ray. In addition, an occupational exposure must be known. As HW with regular contact to infectious patients are examined every year or every third year and all other HW are examined after contact to an infectious patient or infectious materials, the assessment of the occupational exposure is seldom a problem. 

We also conducted a retrospective survey in December 2019 to study both the progression rate and the use of TPT among HW. Workers insured by BGW with an LTBI recognised as an occupational disease between 2009 and 2018 were invited to participate in the survey. The participants received a standardised questionnaire to complete themselves. We wrote to a total of 1709 insured patients. Due to incorrect addresses, 349 insured persons (20.4%) could not be contacted. The questionnaire was returned by 732 persons, the adjusted response rate was 53.8%. Data analysis comprised 575 questionnaires (78.6%). We exclude 149 questionnaires (20.4%) due to implausible responses. Moreover, questionnaires were completed by eight relatives for deceased insured patients (1.1%) and by one relative for an insured patient with dementia. Because the relatives did not provide complete information, these eight questionnaires were also excluded from the analysis. 

The questionnaire was developed for the study and reviewed in a pre-test with ten test subjects to ensure that it was understandable. The questionnaire contained sociodemographic questions, as well as questions about the diagnosis of the LTBI, TPT, and the development of active TB. 

For continuous variables, the mean value was calculated with the standard deviation (SD) and the interquartile range (IQR). Categorical variables were expressed as absolute and relative frequencies. Binary logistic regression with the use of TPT as a target value was used to study influencing factors. Odds ratios (OR), the 95% confidence intervals and p-values were also calculated. The respondents’ professions were grouped into physicians, activities primarily involving patient contact (e.g., nurses) and activities with little or no patient contact (e.g., domestic services staff). The data analysis was performed using SPSS version 26 (IBM Corp., Armonk, NY, USA). The significance level was set at 5%. 

## 3. Results

For the questionnaire analysis, 575 insured patients were eligible for inclusion, which represented 42.3% of all insured patients who received a questionnaire. The study population predominantly consisted of women (82.6%), aged 46.4 years old on average at the probable time of infection (SD 11.1; IQR 16.0). The observation time after diagnosis of an LTBI was one to three years in 32.2% of cases, four to six years in 32.5% of cases, and seven to twelve years in 35.3% of cases. The majority of respondents (62.8%) were working in a hospital at the time of exposure. Respondents were most frequently employed in healthcare and nursing (42.4%). The LTBI was diagnosed for most respondents by company physicians (57.5%) (Table 2).

One-third (33.2%) of survey respondents reported having undergone TPT (Table 3). The most common treatment method was isoniazid over a period of nine months (45.0%). Shorter treatment regimens using rifampicin were performed rarely; rifampicin was mostly administered in combination with isoniazid (3.7%). The most common reason given for use of therapy was an existing immunodeficiency (10.6%). Other reasons stated were autoimmune and tumor-related diseases as well as kidney diseases requiring dialysis.

The use of TPT was dependent on age (Table 4). In the under-30 age group, 52.7% reported having undergone TPT. In the 40–49 and 50–59 age groups, use of TPT was significantly lower at 34.6% and 24.9% respectively. Compared to physicians, health workers in activities primarily involving patient contact (OR 2.6 (95% CI: 1.3 to 5.0)) and health workers in activities with little or no patient contact (OR 3.1 (95% CI: 1.6 to 6.3)) more frequently underwent TPT. The year of diagnosis, gender, and reason for screening all had no significant impact on the use of TPT (Table 4).

A total of six insured patients with progressive cases were identified (Table 5). No progressive cases were found by means of LTBI/TB pairing in the BGW occupational disease routine database. Treatment costs exceeding EUR 7000 were incurred in 16 cases, three of which we identified as progressive cases. Table 5 provides a description of these three cases. The remaining high-cost LTBI cases were not related to progression to TB.

In the survey, another three insured patients reported an LTBI progressing to active TB (Table 6). The three progressive cases were women who worked in different medical fields at the time of infection. All of them had undergone TPT, and two of them had risk factors for progressive LTBI. For two cases, the time between the LTBI and TB diagnosis was less than one year; and for the third person there were two years between the LTBI and TB diagnoses.

The cumulative TB incidence in the retrospective survey cohort was 0.52% (95% CI: 0.14% to 1.65%) (Table 7). Taking into account the time at risk totaling 3080 person years, the incidence was 0.97 cases per 1000 person years (95% CI: 0.25% to 3.10%).

## 4. Discussion

The data presented here describe, for the first time, the annual progression rate from LTBI to active TB and the use of TPT among HW in Germany. One-third of HW with LTBI underwent TPT. The most common indication reported by the insured patients for TPT was immunodeficiency, so treatment seems to be conducted on the basis of risk. Within the overall cohort of 1711 health workers who had received recognition of LTBI as an occupational disease between 2009 and 2018, we identified six HW whose LTBI was progressive (0.4%). For the HW subgroup that participated in the survey (making it the most valid available dataset), the cumulative incidence of progression was lower than generally expected at 0.52%. The incidence density was one case per 1000 person years. 

Compared to the general population with an estimated progression rate of around 5 to 10% within the first two years of infection, our study shows a lower risk of progression [6]. One possible explanation is the “healthy worker” effect, and the lower occurrence that this entails of risk factors that favour progression, such as alcoholism, nicotine consumption or intravenous drug use [11,14,15]. The reason for performing the diagnosis was an examination by a company physician in the majority of cases (56.5%). In addition to occupational check-ups (as required by the Ordinance on Preventive Occupational Medicine) following TB contact, examinations by company physicians may also be conducted in connection with new employment or in mandatory regular examinations due to the occupational risk that arises from the care and treatment of patients with tuberculosis. This means that some infections from further in the past might have been identified that already had a low progression risk at the time of diagnosis. 

Results from the USA and Canada regarding the use of TPT are inconsistent and range from 20 to 80% [16,17,18]. In our study, one-third of the participants underwent TPT. As in previous studies, the use of TPT varied between physicians, HW in activities primarily involving patient contact, and HW in activities with little or no patient contact [9,19]. The medical training and knowledge of HW, for example, regarding side effects, may be a factor potentially discouraging them from undergoing TPT. Moreover, TPT is not recommended in Germany for patients aged 50 or over [8]. It is noteworthy that only one third of the participants answered questions concerning reason for and against TPT (Table 3). This might be because they were not well aware of the pros and contras of TPT. Therefore, better information about progression risk and indications for TPT for HW seem warranted. 

In the USA, the guidelines on the testing, screening and treatment of TB were updated in 2019. Health workers with a diagnosed LTBI are recommended to undergo TPT, unless there are prior contraindications (such as previous LTBI treatment) [20]. Our study suggests that the progression rate for health workers is likely to be low. It thus seems advisable to define case-specific indications for TPT. However, further studies are needed to support this conclusion. 

Although five cases progressed to active TB despite TPT, this remains an effective method of mitigating the risk of progression. It should therefore be assumed that the progression rate would have been higher without the use of TPT in this study population. Our data show that risk groups should continue to be observed even after TPT. 

In terms of limitations, it should be noted that it was impossible to verify the results due to the anonymous nature of the survey. Moreover, the progression risk may be underestimated by non-participation in the survey or the early death of TB patients. Given the generally low mortality of TB patients in Germany, however, the latter is considered to be unlikely [21]. There is also the possibility that insured patients have not reported active TB progression to the BGW. In addition, we know the reporting date of the LTBI, but not the infection period. Where infections occurred further in the past, it is possible that progressive cases were not identified by our study, which may also have led to underestimation of the progression rate.

It should be borne in mind that TB is not the only occupational risk for HW. COVID-19 made evident how vulnerable HW are worldwide and in Germany [22,23]. Blood-borne virus infections are another risk. However, prevention, safe instruments, vaccination and new treatments helped to reduce the risk for HW [24]. Besides infections, skin diseases and low back pain are other prevalent work-related diseases in HW, which warrant prevention and health promotion [25]. 

## 5. Conclusions

The occupational check-ups offered for health workers following contact with TB patients in low-incidence countries such as Germany are an important instrument for TB prevention in healthcare. They ensure that health workers are informed about the risks of TB infection and enable the early detection of TB. Following our data, it seems warranted to better inform HW about the option and the benefit–risk profile of TPT. As most HW who progressed from LTBI to active TB had medical risk factors for TB, the administration of TPT seems most needed in HW with pre-existing health problems.

## Figures and Tables

**Table 1 ijerph-18-07053-t001:** Recognised occupational latent tuberculosis infections (LTBI). Data from the German Statuary Institution for Accident Insurance and Prevention for Health and Welfare Services (BGW) for 2009 to 2018 (*n* = 1711).

Year	Number of LTBI Recognitions
2009	67
2010	126
2011	171
2012	181
2013	241
2014	204
2015	180
2016	159
2017	202
2018	180
Total	1711

**Table 2 ijerph-18-07053-t002:** Description of the study population from the survey (*n* = 575).

	Characteristic	*n*	%
Gender	Women	475	82.6
	Men	100	17.4
Age groups at time of diagnosis	Under 30	55	9.6
	30–39	91	15.8
	40–49	162	28.2
	50–59	209	36.3
	≥60	58	10.1
Duration of observation, grouped	1–3 years	185	32.2
	4–6 years	187	32.5
	7–12 years	203	35.3
Occupational field at time of tuberculosis exposure	Hospital ^1^	361	62.8
	Dental/medical practice	82	14.3
	Inpatient, outpatient care services	67	11.7
	Social and advisory services	36	6.3
	Laboratories	9	1.6
	Other ^2^	20	3.5
Occupation at time of tuberculosis exposure	Healthcare and nursing specialist	244	42.4
	Physician	82	14.3
	Specialist medical assistant	77	13.4
	Specialist medical technical assistant	39	6.8
	Nursing assistant	30	5.2
	Geriatric nurse	24	4.2
	Social work	23	4.0
	Other ^3^	56	9.7
Reason for tuberculosis screening	Preventive check-up company physician	325	56.5
	Screening public health authority	185	32.2
	Other ^4^	65	11.3

^1^ Including rehabilitation clinics. ^2^ Dialysis facility, public health authority, pathology, psychiatric institution, school, administrative institution, no response. ^3^ Trainee, occupational therapist, hairdresser, domestic services staff, remedial therapist, managerial function, teacher, patient support service, physiotherapist, psychologist, autopsy technician. ^4^ Assignment abroad, intention to receive BCG vaccination, clinical symptoms, needlestick injury, rheumatism diagnosis, routine examination as required by the Ordinance on Preventive Occupational Medicine, study participation, examination by regular general practitioner, elective precautionary measure.

**Table 3 ijerph-18-07053-t003:** Information provided on TPT (*n* = 191; 33.2%) from survey.

TPT	Characteristic	*n*	%
Treatment regimen	Isoniazid, 6 months	68	35.6
	Isoniazid, 9 months	86	45.0
	Isoniazid, rifampicin, 3 months	7	3.7
	Other ^1^	23	12.0
	Unknown (drug or time period)	7	3.7
Reasons stated for undergoing TPT ^2^	Immunodeficiency	19	10.6
	Rheumatism treatment	6	3.4
	Kidney disease/dialysis	1	0.6
	Tumours	2	1.1
	(Planned) pregnancy	5	2.8
	Other reason ^3^	26	14.5
	Unknown	120	67.0
Reasons stated against use of TPT ^2,4^	Aged over 50	102	20.3
	No immunodeficiency	203	40.4
	Unknown time of infection	62	12.3
	Existing liver disease	7	1.4
	Other reason ^5^	48	9.5
	Unknown	81	16.1

^1^ Isoniazid, 3 months; isoniazid, 4 months; isoniazid, 12 months; isoniazid and rifampicin, 4 months; isoniazid and rifampicin, 6 months; rifampicin, 4 months; rifampicin, 6 months. ^2^ Multiple responses possible. ^3^ Young age, underlying disease (of lungs), therapy recommended by public health authority or attending physician. ^4^ Reference value in this case: *n* = 384. ^5^ Personal rejection, (planned) pregnancy, no therapy recommendation expressed by public health authority or attending physician, drug allergy or expected side effects, multi-resistant pathogen in carrier.

**Table 4 ijerph-18-07053-t004:** Factors influencing the use of TPT from the surveys (*n* = 575).

Influencing Factors	*n*	TPT		% Yes (**)	OR	(95% Confidence Interval)
		Yes	No			
Year of diagnosis						
2016–2018	184	59	125	32.1		
2013–2015	188	59	129	31.4	0.9	(0.6–1.4)
Before 2013	203	73	130	36.0	1.2	(0.7–1.8)
Gender						
Female	475	162	313	34.1		
Male	100	29	71	29.0	1.1	(0.7–1.9)
Age upon infection						
Under 30	55	29	26	52.7		
30–39 years	91	35	56	38.5	0.6	(0.3–1.3)
40–49 years	162	56	106	34.6	0.5 *	(0.3–0.9)
50–59 years	209	52	157	24.9	0.3 *	(0.2–0.6)
Over 60	58	19	39	32.8	0.5	(0.2–1.1)
Activity						
Physician	82	15	67	18.3		
Activity involving patient contact	312	105	207	33.7	2.6 *	(1.3–5.0)
Activity with little or no patient contact	181	71	110	39.2	3.1 *	(1.6–6.3)
Reason for screening						
Preventive check-up by company doc	325	103	222	31.7		
Screening by public health authority	185	63	122	34.1	1.0	(0.7–1.5)
Other	65	40	25	61.5	1.3	(0.7–2.3)

* Significant. (**) Expressed as row percentage.

**Table 5 ijerph-18-07053-t005:** Case summary of progressive cases based on file analysis.

Progressive Cases from Case File Analysis			
	Case 1	Case 2	Case 3
Age at time of LTBI diagnosis	51 years	24 years	39 years
Gender	Female	Male	Female
Activity	Inpatient care	Homelessness assistance	Outpatient care
TPT	Yes	Yes	Rejected by patient
Indication for TPT	Immunosuppressant medications	Recommended due to young age	Unknown
TPT regime	Isoniazid/9 months	Isoniazid/6 months	
Progression after	Three years	Eighteen months	Ten months

**Table 6 ijerph-18-07053-t006:** Case summary of progressive cases based on survey.

Progressive Cases from Survey			
	Case 1	Case 2	Case 3
Age at time of diagnosis	52 years	28 years	27 years
Gender	Female	Female	Female
Activity	Inpatient care	Medical technical assistant	Specialist medical assistant
TPT	Yes	Yes	Yes
Indication for TPT	Rheumatic diseases	Immunodeficiency	Unknown
Treatment regime	Isoniazid/9 months	Isoniazid/6 months	Isoniazid/6 months
Progression after	Two years	<one year	<one year

**Table 7 ijerph-18-07053-t007:** Observation period and progression rate among 575 workers with occupational LTBI with code 3101 from the survey.

*n*	Observation Period in Years				TB	Cumulative Incidence (95% CI)	Incidence per 1000 Person Years (95% CI)
	Mean	SD	Median	IQR			
575	5.4	2.8	5.0	5.0	3	0.52% (0.14%–1.65%)	0.97 (0.25–3.10)

## Data Availability

Data of the study are available upon request by the authors.

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
