# Peer review of "Latent Tuberculosis Infection among Health Workers in Germany—A Retrospective Study on Progression Risk and Use of Preventive Therapy"

_ijerph, 2021, doi:10.3390/ijerph18137053_

Round 1
Reviewer 1 Report
General comments
This was a well conducted and straight forward study. It was also easy reading through.
In the abstract, the authors should mention what instrument was used to collect the information. Was the data collected from a database?
Specific comments
Line 22-23: In Line 22, the authors say the study was conducted among 1,711 HW, and in Line 23, they say the included 575 HW. This is confusing and needs to be rephrased.
Line 46: German Statutory Institution?
Line 50” Italicise scientific names
Lines 87-90: Please do not start sentences with numbers
Lines 96-97: … questions about the diagnosis of the LTBI, preventive chemotherapy, and the development of active TB.
Line 111:… women (82.6%), aged 46.4 years old on average….
Line 117-118: If the authors analysed 575 full responses, then it is erroneous to calculate the number of HW who progressed to active TB as a percentage of 1,711. It is possible that if all the 1,711 patients responded, the progression rate might have been different.
Author Response
We are thankful for the thoughtful comments of the reviewer. We made changes accordingly as can be seen in our one-to-one response.
Reviewer 1
This was a well conducted and straight forward study. It was also easy reading through.
1. In the abstract, the authors should mention what instrument was used to collect the information. Was the data collected from a database?
Response of authors.
Thank you for the comments. We added in the abstract that a self-administered questionnaire was used for the retrospective survey
Specific comments
2. Line 22-23: In Line 22, the authors say the study was conducted among 1,711 HW, and in Line 23, they say the included 575 HW. This is confusing and needs to be rephrased.
Response of the authors
Thank you for the comment. We rephrased the sentence and made two sentences out of it. Now I should be easier to understand that we analysed two different datasets.
Routine data from the German Statutory Institution for Accident Insurance and Prevention for Health and Welfare Services (BGW) was analysed and a retrospective survey was conducted. A self-administered questionnaire was send to 1,711 HW who had received recognition of a LTBI as occupational disease between the years 2009 and 2018. The response rate was 42.3% after correcting for those with no actual address (20.4%). We included 575 HW in the data analysis of the retrospective survey.
3. Line 46: German Statutory Institution?
Response of authors
Thank you. Spelling was corrected.
4. Line 50” Italicise scientific names
Response of authors
We changed it accordingly
5. Lines 87-90: Please do not start sentences with numbers
Response of authors
Thank you for the comment. The sentences were rephrased. Now we write:
Due to incorrect addresses, 349 insured persons (20.4%) could not be contacted. The questionnaire was returned by 732 persons, the adjusted response rate was 53.8%. Data analysis comprised 575 questionnaires (78.6%). We exclude 149 questionnaires (20.4%) due to implausible responses.
6. Lines 96-97: … questions about the diagnosis of the LTBI, preventive chemotherapy, and the development of active TB.
Response of authors
Thank you for the comment. We deleted about
7. Line 111:… women (82.6%), aged 46.4 years old on average….
Response of authors
We changed it accordingly
8. Line 117-118: If the authors analysed 575 full responses, then it is erroneous to calculate the number of HW who progressed to active TB as a percentage of 1,711. It is possible that if all the 1,711 patients responded, the progression rate might have been different.
Response of authors
We agree with the reviewer. The progression rate was calculated using 575 patients as reference.
Reviewer 2 Report
It is recommended to add specifics about the diagnosis of LTBI, such as the method of testing, when and how often it is applied.
Author Response
We are thankful for the thoughtful comments of the reviewer. We made changes accordingly as can be seen in our one-to-one response.
Reviewer 2
It is recommended to add specifics about the diagnosis of LTBI, such as the method of testing, when and how often it is applied.
Response of authors
Thank you for the comments. We made amendments accordingly:
LTBI is diagnosed by performing an immune test (TST or IGRA) and by excluding active TB via chest X-ray, when the immune test is positive. HW with regular contact to infectious patients are examined every year or every third year. Al other HW are examined after a contact to an infectious patient or infectious materials.
Reviewer 3 Report
Zielinsky et al., described information about latent tuberculosis infection among health workers. This type of survey can be useful for health care workers especially for those who have direct or indirect contact with potential infection. To help improve the manuscript, I made specific comments listed below.
1. WHO standard LTBI antibiotics treatments such as isoniazid, rifampin were treatment regimens according to the table 3. Why use the word "chemotherapy"? I suggest changing to antibiotics.
2. Table 1: Add total number in the table since text only describes total number.
3. How "probable the time of infection" was determined? Can authors provide more data how long days, weeks or months individuals were exposed to potential TB, and from how many TB infected peoples, etc? It would be valuable data to add to the current manuscript so that the readers can have a better understanding of what kind of TB exposure these health workers are dealing with.
4. Table 5 has two tables merged together therefore it is difficult to read. Can authors rearrange table 5?
5. According to Table 4, the majority of people didn't undergo preventive treatment. Can authors provide additional data about why and discuss what we can do about it?
Author Response
We are thankful for the thoughtful comments of the reviewer. We made changes accordingly as can be seen in our one-to-one response.
Reviewer 3
Zielinsky et al., described information about latent tuberculosis infection among health workers. This type of survey can be useful for health care workers especially for those who have direct or indirect contact with potential infection. To help improve the manuscript, I made specific comments listed below.
1. WHO standard LTBI antibiotics treatments such as isoniazid, rifampin were treatment regimens according to the table 3. Why use the word "chemotherapy"? I suggest changing to antibiotics.
Response of authors
Thank you for the comment. We changed chemotherapy to Tuberculosis Preventive Treatment (TPT) in accordance with WHO.
2. Table 1: Add total number in the table since text only describes total number.
Response of authors
Thank you, good point. We added the total number
3. How "probable the time of infection" was determined? Can authors provide more data how long days, weeks or months individuals were exposed to potential TB, and from how many TB infected peoples, etc? It would be valuable data to add to the current manuscript so that the readers can have a better understanding of what kind of TB exposure these health workers are dealing with.
Response of authors
Thank you for the comment. In detail this data are not available. However, we know the indications for testing. From this, the exposure situation can be deduced. This in now described in the method part. See also comment of reviewer 2 and the response of the authors.
4. Table 5 has two tables merged together therefore it is difficult to read. Can authors rearrange table 5?
Response of authors
Thank you for the comment. We turned Table 5 into two Table (Table 5 and Table 6)
5. According to Table 4, the majority of people didn't undergo preventive treatment. Can authors provide additional data about why and discuss what we can do about it?
Response of authors
Thank you for the comment. In table 3 we provide information on reasons for and against TPT. Unfortunately, only one third of the participants provides answers. This might be because they are not well informed about the option of a TPT. Therefore, a better information about the pro and cons of TPT might be needed. Now we mention this in the discussion.
Reviewer 4 Report
1. This research comes at a very important time of covid-19 where discussion about occupational hazards for health professionals are critical. Although this is a different occupational hazard but the relevance is similar and hence Authors need to say something in the introduction about this context . This can be even one line.
2. Results in table one are presented in one of the most boring modes for the reader. Even a line graph would have done a better job. A table is certainly not the best
3. Line 92 - You cannot start a sentence in an academic paper with ...."because..."
4. The first sentence of the conclusion is too long. Please rework this
5. Please strengthen your conclusion
6. Note that the discussion was done very well except that there was no reference to other occupational diseases
Author Response
We are thankful for the thoughtful comments of the reviewer. We made changes accordingly as can be seen in our one-to-one response.
Reviewer 4
1. This research comes at a very important time of covid-19 where discussion about occupational hazards for health professionals are critical. Although this is a different occupational hazard but the relevance is similar and hence Authors need to say something in the introduction about this context . This can be even one line.
Response of authors
Thank you for this comment.
Covid-19 and TB are both airborne diseases. As the infection-risk of HW for TB is well established there can be lessons learnt from TB prevention for COVID-19 prevention. However, it needs to be emphasized that the number of HW with COVID-19 is much higher than the number of HW with LTBI as an occupational disease. In Germany, during the last 13 month more than 50,000 occupational diseases because of COVID-19 were recognised, while about 200 occupational diseases because of LTBI are recognized every year.
2. Results in table one are presented in one of the most boring modes for the reader. Even a line graph would have done a better job. A table is certainly not the best
Response of authors
We agree with the reviewer. However, as reviewer 3 suggested to add the total to the table, we prefer to stay with the table (see comment 2 of Reviewer 3).
3. Line 92 - You cannot start a sentence in an academic paper with ...."because..."
Response of authors
Thank you for the comment. We rephrased the sentence accordingly
4. The first sentence of the conclusion is too long. Please rework this
Response of authors
We rephrased the first sentence. See also comment 4.
5. Please strengthen your conclusion
Response of authors
Thank you for the comment. We strengthened the conclusion by saying that HW need to be better informed about TPT. Now the conclusion is as follows:
The occupational check-ups offered for health workers following contact with TB patients in low-incidence countries such as Germany are an important instrument for TB prevention in healthcare. They ensure that health workers are informed about the risks of TB infection, and enable the early detection of TB. Following our data it seems warranted better informing HW about the option and the benefit-risk profile of a TPT. As most HW who progressed from LTBI to active TB had medical risk factors for TB, the administration of TPT seems most needed in HW with pre-existing health problems.
6. Note that the discussion was done very well except that there was no reference to other occupational diseases
Response of authors
Thank you for the comment. Now we include a reference to other occupational diseases:
It should be borne in mind that TB is not the only occupational risk for HW. Covid-19 made evident how vulnerable HW are worldwide and in Germany [22, 23]. Another risk are blood borne virus infections. However, prevention, safe instruments, vaccination and new treatments helped to reduce the risk for HW [24]. Besides of infections, skin diseases and low back pain are other prevalent work related diseases in HW, which warrant prevention and health promotion [25].
Round 2
Reviewer 3 Report
Authors responded to each comment and made proper improvement.